# The Nexus of Sports-Based Development and Education of Mental Health and Physical Fitness

**DOI:** 10.3390/ijerph20043737

**Published:** 2023-02-20

**Authors:** Tiejun Zhang, Huarong Liu, Yi Lu, Qinglei Wang

**Affiliations:** 1Sports Ministry, Henan University of Economics and Law, Zhengzhou 450011, China; 2School of Physical Education, China University of Geosciences, Wuhan 430074, China; 3Sports Teaching and Research Section, Wuhan University of Communications, Wuhan 430205, China; 4Faculty of Sport and Exercise Sciences, University of Malaya, Kuala Lumpur 50603, Malaysia

**Keywords:** physical components, mental components, mental health, intervention, physical activity

## Abstract

Physical inactivity has increased globally, particularly in developed nations. A high proportion of the human population is unable to meet the physical activity recommendation of the World Health Organisation due to hypertension, metabolic syndrome, obesity, and other medical conditions. Non-communicable diseases and mental health problems are becoming more prevalent, particularly in low and middle-income nations. This study aimed to determine the effectivenessof a mentorship programmeon university students’ mental health and physical fitness. The intervention comprised the effects of sports-based development and education on physical fitness and mental health. A total of 196 and 234 students from two universities were randomly assigned to the intervention and control groups, respectively. The primary outcomes were engagement in physical activities (number of push-ups for 1 min, the strength of hand grip (kg), and the Jump test while standing (cm)), body fat proportion and psychological resilience, self-efficacy, and relationship with family and schoolmates. Participants in the control group had access to a web-based health education game, whereas the intervention group wassubjected to intensive interventional activities for one month based on the eight principles of the National Research Council and Institute of Medicine. Data were analysed using Analysis of Variance (ANOVA) to compare the physical and mental components between the intervention and control groups. Relative to baseline, all the physical health components (push-ups, sit-ups, and jump tests), psychological resilience, relationship with family members, and self-efficacy increased significantly in the intervention compared to the control group. Body fat composition was significantly reduced in the intervention when compared tothe control group. In conclusion, the mentorship programme effectively improved the participants’ physical and psychological health and could be developed further for application in a larger population.

## 1. Introduction

Physical activity is defined as “any bodily movement that occurs with energy consumption using our muscles and joints” [1]. Physical activity contributes significantly to the development of physical and spiritual aspects of individuals by promoting community well-being, protecting the environment, and investing in future generations. Meanwhile, physical inactivity remains a global health problem [2,3].

Physical inactivity is a common risk factor for chronic diseases [4]. It increases the odds of non-communicable diseases, such as diabetes, obesity, high blood pressure, cancers and metabolic diseases. In other words, physical inactivity plays a vital role in the reduction in life expectancy and quality, while being among the highly-ranked risk factors for mortality worldwide (6.0% of deaths globally) [5,6].

Specifically, physical inactivity is considered the main cause of 20–25% of colon and breast cancers, 30% of ischaemic heart disease, and 27% of diabetes [6,7]. The increased prevalence of obesity has also been linked to widespread sedentary lifestyle and physical inactivity [6]. These events are supported by findings from epidemiological studies that showa 30–50% increased prevalence of major causes of mortality occurring in the physically inactive group [2]. Figure 1 presents the common consequences of physical inactivity and its link to psychologically or biologically stimulating insidious bad habits, such as low-function fitness, smoking, poor nutrition, and psycho-social distress, as highlighted by [8]. These factors may eventually increase the risk of serious illnesses such as diabetes, cardiovascular disease, stroke, kidney disease, and various cancers.

Mental health issues are now a major concern in low- and middle-income countries (LMICs). The incidence of mental disorders is increasing at an alarming rateas more people are moving to cities with and are therefore being exposed to several risk factors fordeveloping anxiety [10]. Physical inactivity is also associated withurban development and the rise of health complications globally [11,12]. As unplanned urbanisation increases, the availability of open spaces decreases, leading to a decline in physical activity [12]. A Cochrane review of PA and health wellness benefits in school-aged youngsters revealed that urbanicity positivelyimpacts risk factors for non-communicable diseases and psychological health [13].

The recent COVID-19 pandemic has also heightened the risk of mental disorders, as factors such as social isolation—studying and working at home, reduced physical interaction and contact with other individuals may be strong psychological stressors [14,15]. These factors elicit negative lifestyle alterations such as poor diet and physical activity [16]. The lack or reduction in physical activity has been associated with a higher risk of mental disorders, including depression and anxiety [17]. In contrast, a positive relationship was reported between improved psychological outcomes and physical exercise [18], as its neurobiological impacts appear to influence diverse neural mechanisms relating to anxiety and depressive disorders [19]. Despite there being no ideal dose of exercise for mitigating mental disorders that hasbeen documented, accumulated evidence from the literature suggests that the risk of mental disorders is reduced by any exercise compared to complete physical inactivity [17,18]. Aprior study also indicatedthat regular physical exercise is comparable to pharmacological measures for the management of depression and anxiety [20].

Mental disorders and other consequences of physical inactivity are profound in children, especially during adolescence [21]. Several studies have documented the benefits of a physically active lifestyle during adolescence, ranging from better muscular fitness and cardiometabolic and cardiorespiratory health to positive effects on BMI and body weight [22,23]. Additionally, improved prosocial behaviour and cognitive development among children have been linked to the positive impacts of physical activity [1]. To achieve these benefits, the WHO and US guidelines recommended that adolescents should perform at least 60 min of physical activity daily [24,25]. As a result, recent studies have advocated for interventions to promote engagement in physical activity and lifestyle behaviours to improve the mental and physical health of school-aged children and the adolescent population [26].

Overall, physical exercise appears to offer benefits in reducing anxiety and depression scores in children and adolescents, but it is difficult to draw firm conclusions due to the limited number of research findings, the clinical diversity of participants, interventions, and measurement methods [20]. In the past 15 years, the number of research studies on adolescent major depressive disorder due to physical inactivity has increased dramatically, with a significant proportion being clinical studies of medication and cognitive behavioural therapy. However, the response and remission rates have been low [27]. Additionally, the majority of positive responders post-treatment experienced many persistent symptoms, in addition to seriously impaired functioning, and high levels of relapse. More efficient therapies are required to cure this severe and chronic illness that typically lasts into adolescence and haspoor long-term consequences. Early treatment studies indicated that exercise and other therapies might effectively treat teenage major depressive disorder [28]. There are few robust analyses of existing sport-based development initiatives, and there is little proof that backs up such claims [13]. For this reason, research on evaluating dataneeds to be carried out to address these gaps.

It is critical to ascertain whether reducing depressive symptoms may be achieved with a non-medication treatment, especially in light of recent concerns regarding the long and short-term safety of selective serotonin reuptake inhibitors (SSRIs) and other antidepressants [29]. Mentorship programmes have been considered to be an effective non-pharmacological approach to ameliorating depressive symptoms and other health problems linked to physical inactivity among school-age children [30]. For instance, peer-to-peer mentoring provided by local high school students strengthened sustainable behavioural change and had positive impacts on BMI and engagement in daily physical activity [29]. However, the effectiveness of mentorship regarding sport-based development on physical and mental health has not been widely explored. This study aimed to determine the efficacyof a mentorship programme based on sport-based development on university students’ physical and mental health.

Reviews of the Intervention Impacts on Physical Fitness and Mental Health

The sport-for-development intervention was the Gum Marom Kids League (GMKL), a society programme that ran for 11 weeks [31]. The primary objective of the GMKL was to employ sports to enhance peoples’ physical and mental health, subsequently making the neighbourhood a more enjoyableplace [32,33]. During the GMKL, all participants performed significantly better in all activities on Mature Fine Tailing (MFT). However, for men and women, the between-group MFT treatment effects were slightly different but not statistically significant [28]. According to the girls’ data after adaptation, individuals who received assistance fared much better than the group that did not receive any assistance [34].

The sport-for-development infrastructure positsthat sports benefit society and health [35]. Evidence shows that these arguments are not always accurate and contradict the statistical inferences that PA benefits young people’s mental well-being [32,34]. This is crucial to emphasise for governments and organisations that manage initiatives, develop policies, and engage with patients to disseminate these alerts and encourage participation in sports to increase PA and enhance health [36]. One optimistic way to look at the mental health outcomes is to assume that the adolescents involved directly in the sport-for-development intervention felt more at ease interacting with their challenges, which led to a role conflict. However, the instrument used to assess mental health was responsive, valid, and efficient for analysing depression and anxiety-like illnesses [36,37].

## 2. Materials and Methods

### 2.1. Research Design

This study was a randomised control trial involving an experimental (intervention) and a control group (no intervention). The study was conducted in Henan province, China among university students aged 17–28 years old from the Henan University of Economics and Law and Zhengzhou University, Henan province, China. In total, 430 participants were recruited and randomly assigned to either the intervention or the control group. Randomisation was performed by tossing a coin. Participants in the intervention group participated in a sports mentorship programmefrom May 2022 to June 2022.

### 2.2. Inclusion and Exclusion Criteria

This study included students from the two aforementioned universities who were between the ages of 17 and 28 years old. Students who agreed to follow the whole study process and were interested in sports were included. All students provided written consent and participation was voluntary. The students who had underlying chronic disorders, diabetes, respiratory or cardiological abnormalities, and a history of mental disorders and those who opted out in the middle of the study were excluded from this study.

### 2.3. Recruitment and Randomization

Since this study was conducted atthe researchers’ institutions, it was easy to contact the potential participants in the relevant departments for recruitment purposes. The head of the department of the Sports Ministry was contacted and briefed on the research objectives and procedures. Thereafter, the students’ contact information was obtained from the relevant office in the department. Students were also reached during face-to-face classes to inform them of the research objectives, procedures and benefits. Upon applying the inclusion and exclusion criteria, a total of 400 students were considered eligible to participate in the study.

Randomisation was performed by tossing a coin. Given the sampling frame of 400 students, a participant was randomly assigned to the control group upon obtaining a head from tossing a coin and the next student was allocated to the intervention group. This process led to a total of 236 and 194 participants in the control and intervention groups, respectively.

### 2.4. Ethical Approval

The Ethical Committee of the Henan University of Economics and Law, and Zhengzhou University, Henan province, China, approved this study. The ethical approval was assigned as 202203/HNCN1003/168. Each participant was briefed about the study process and objectives, and the students were informed of their respective groups (either intervention or control) before starting the intervention. Participation was voluntary and only those that provided written and informed consent were recruited for the study. Apart from the expected benefits of participating in the intervention, no incentive was given for participating in this study.

### 2.5. Intervention

#### 2.5.1. Framework

This afterschool sports mentorshipcontrastswithphysical education programmesthat emphasise sporting abilities, use sports to empower adolescents, and foster life skills. The eight principles of Positive Youth Development (PYD) are sports skills, school engagement, healthy living, positive character, self-direction, teamwork, leadership, and community engagement. These are derived from the National Research Counciland the Institute of Medicine, and both served as the foundation of the intervention framework. This framework was adapted from theabove institute so that the principles would be valid and modified to suit the Chinese population. These principles guaranteed the students’ physical and mental wellbeing, a friendly and productive environment (such as supportive relationships and constructive social norms), and an appropriate programme framework for skill development. This framework was expected not to affect the participants’ mental status.

#### 2.5.2. Delivery of the Intervention

The participants (aged 17–28 years old) received the intervention in small groups, which made it convenient to organizeeach day’s schedule and monitor the participants. This intimate setting provided a chance for interaction, a feeling of community, and a more supportive and youth-centred environment. Each small group of 17 students selected the sport they wanted to learn. Among the sports chosen by the students were kickboxing, basketball, and volleyball, which were based on their availability at the University. This programme adopted a different approach compared to traditional physical education sessions and was not teacher-centred. Instead of being a teacher, the mentors insteadserved as a facilitator. For instance, no established curricula specified what sporting skills should be taught. Each group’s learning routes were chosen through conversation between the mentors and the students.

The intervention was delivered by mentors, who were certified sports coaches from regional sports organisations. In addition, they received instructions on howto use PYD ideas and sports psychology throughout their prior professional training (higher diploma or bachelor’s degree). Before the intervention, the study team provided a one-day session covering PYD and youth psychology basics, safety measures for physical and mental health, the program’s guiding principles, and a semi-structured curriculum. Anenvironment that prioritises young people was emphasised. As an illustration, the mentors were instructed to assist the participants in establishing their athletic goals. Additionally, problem-solving methods specific to the world of sports were offered. These were created to encourage the development of resilience.

#### 2.5.3. Structure and Components of the Intervention

The curriculum was the same for each group and was semi-structured. The intervention was split into two comparable halves (each with nine sessions), spaced apart by school holidays and examinations in December and January. Each section beganwith two sessions of warm-up and introduction, during which the mentor presentedthe selected sport through purposeful play. The students then spoke about what kind of sporting goal they would like to achieve over the following half session (45 min). After establishing their goals, the students worked on improving their sporting abilities throughout seven sessions with the help of their peers and mentors, who also provided problem-solving strategies through experiential learning. The physical activities in the programme includedpush-ups for 1 min, the strength of hand grip (kg), and a jump test while standing (cm). Participants were also instructed to rate their relationship with family and schooling activities. Self-efficacy and psychological resilience were also assessed using a structured questionnaire.

The 45-min debriefing that followed was used to consolidate skills and engage in self-reflection. The participants could also use the abilities they had learnedin other areas of their lives. Approximately 83% (or 1350 min) of the programme was allotted for participating in sports or PA, while 17% (or 270 min) was devoted to mentor-led instruction, presentation, and discussion. The majority of the programme was taught in classrooms. When the school could not provide the necessary location or facilities, nearby community centres were utilised.

#### 2.5.4. Monitoring of the Intervention and Outcomes

Researchers and research assistants visited one quarter of the randomly chosen sessions to verify programme fidelity. The mentors were reminded if the observers noticed students who were not participating in the sessions. The testing revealed that the mentors could follow the semi-structured curriculum and PYD principles. Metrics were constructed into the physical training timetable so the group members could easily obtain and keep a record of them. All of the chosen school systems were given an evaluation day scheduled with the school principals. Throughout the week before the day set aside for the evaluation, all of the participants were informed about the procedure and given written material fortheir parents or guardians [31]. For every outcome measured, the level of involvement in the guideline and follow-up test methods was unique and individualised [28,31]. During the study period, the students were instructed not to interact with each otherin order to avoid bias.

Meanwhile, the students in the control group were asked to access a web-based health education game with 400 questions on healthy lifestyles during the study period. The university’s computer room was provided to thosestudents without internet or computer access. Similarly, students were not allowed to interact in order to minimise bias.

### 2.6. Assessment of Primary Outcomes

The primary outcomes assessed in this study are broadly categorised into three broad aspects: physical activities, physical and mental health, and relationships/self-efficacy. For physical activities, the variables considered include the overall score for physical activity, number of push-ups for 1 min, the strength of hand grip (kg), and the jump test while standing (cm), whereas body fat proportion and psychological resilience were recorded as indicators of physical and mental health, respectively. Body fat composition was estimated by assessing waist circumference as described by [38]. Prior research has reported significant correlations between waist circumference and measures of abdominal fatness measured by magnetic resonance imaging [39,40]. For the third aspect, self-efficacy and relationships with family and school were documented, as mentioned in the previous section.

### 2.7. Statistical Analysis

This study used SPSS 25 (IBM Corporation, Armonk, New York, Uniteed States) and Excel software for effective statistical analyses. All the data were subjected to normality tests based on the levels of kurtosis and skewness. As a result, all of the data conformed to the assumptions of normality tests and were considered normally distributed. The descriptive measurements were expressed as mean ± standard deviation (SD), and the changes in the parameters were calculated for in-depth analysis before and after the intervention. The study employed ANOVA analysis for the significance test. The level of significance was tested at α = 0.05.

## 3. Results

### 3.1. Descriptive Results

Table 1 summarises the participants’ characteristics in terms of age, gender, BMI and socio-economic class. This result presents the baseline characteristics of each group.

The mean age of participants in the control and intervention groups was 22.52 (±3.60) and 22.29 (±3.15), while the BMI was 26.95 (±2.90) and 26.65 (±2.69), respectively. No significant difference (*p* > 0.05) was recorded for both variables between both groups. In terms of gender, 50.4% (119/236) of the students in the control group were males compared to 48.4% (94/194) in the intervention group. This reflects that male and female participants were randomly assigned to both groups. Similarly, no significant difference was observed between each group in terms of socioeconomic class.

### 3.2. PhysicalActivity and Specific Exercises

Table 2 presents the results for the number of push-ups for 1 min, the number of sit and ups test for 1 min, the strength of hand grip (kg), and overall physical activity. No significant difference was observed in baseline values for each parameter between the intervention and control groups. In the control group, comparisons of each parameter before and after the web-based health education game depicted no significant differences. However, the scores for all the parameters in the intervention group increased significantly post-intervention compared to the baseline values.

### 3.3. Body Fat Composition and Psychological Resilience

As shown in Table 3, body fat composition and psychological resilience did not differ between the groups at the baseline or before the intervention. While no difference was detected in each parameter before and after the web-based health education game in the control group, the intervention group experienced a significant decrease in body fat proportion and increased psychological resilience post-intervention.

### 3.4. Self-Efficacy and Relationship with Family and School Activities

Results for the self-efficacy and vital relationships with family and schoolmates are presented in Table 4. All of the variables were not significantly different between both groups at the baseline. However, the intervention group reflected a significant increase in self-efficacy and relationship with family members after the intervention, but no difference was observed in the relationship with schoolmates. In the control group, none of the parameters was significantly different between pre and post-exposure to a web-based health production game.

## 4. Discussion

This study explored the effectiveness of a mentorship programme in promoting engagement in physical activity among university students at two universities in China. Furthermore, the study investigated the impact of the intervention on specific exercise activities, self-efficacy, body fat composition, psychological resilience, and relationships with family members.

Globally, psychological health and well-being have been emphasised as crucial concerns. As a tool for navigating workplace difficulty, the idea of resilience for professionals has grown in popularity. College students confront unique challenges and are at increased risk for mental health issues. Children and adolescents are frequently at risk for depressive symptoms, including mood swings, apathy, general discontent, guilt, hopelessness, loss of interest, and loss of interest or pleasure in activities [41]. The research on this topic hasintensified (asmentioned above), asunhealthy symptoms are the leadingcause of numerous diseasesand impairments, especially among adolescents and college students.

Several studies have documented the benefits of a physically active lifestyle during adolescence, ranging from better muscular fitness, cardiometabolic and cardiorespiratory health, to positive effects on BMI and body weight [22,23]. Additionally, improved prosocial behaviour and cognitive development among children have been linked to the positive impacts of physical activity [36]. Thus, physical fitness and exercise are one of the most current strategies used to promote youth health outcomes. PAprogrammes and health promotion policies should be created to enhance cardiorespiratory fitness, muscle fitness, and speed agility. Schools may have a significant impact by recognising students with poor physical fitness and encouraging students to engage in healthy activities [29].

The mentorship programme used in this study focused on physical activities developed based on the eight principles of the National Research Council and the Institute of Medicine. Students in the intervention group were exposed to these intensive interventional activities for one month, and vital parameters such as specific exercise activities, self-efficacy, body fat composition, psychological resilience, and relationship with family members were assessed. These parameters were used as indicators of physical and mental health.

In the present study, the intervention group recorded a significant increase in self-efficacy and psychological resilience compared tothe control group. These findings are consistent with prior studies in which exercise performed independently or in conjunction with other measures was found to boost young people’s self-esteem [42]. Similarly, the relevant literaturestated that self-esteem could rise as a result of PA, at least temporarily [43,44]. Children and adolescents frequently experiencepsychological and behavioural issues, and raising one’s self-esteem may help to stop the emergence of such issues. According to the outcomes of previous studies, exercise temporarilyincreases the self-esteem of adolescents and children. This may be owing to the absence of any known detrimental effects and multiple channels for emotional release, which lead to numerous positive benefits on physical health [37]. The increase in psychological resilience experienced in the intervention group implies a strong impact on their physical and mental well-being. Previous studies revealed that resilience played a major role in mediating the link between mental and physical well-being, and encouraging resilient physical activities would be a great strategy for boosting mental health [29,45].

All of the specific exercises investigated in this study (i.e., grip (kg)) and the overall engagement in physical activity improved significantly in the number of push-ups for 1 min, the number of sit and ups test for 1 min, and the strength of the hand in the intervention group compared to the control group. These results suggest that the mentorship programme elicited behavioural change bymotivating students to engage in the physical activities and exercises chosen before the intervention. Similar findings were reported by [30], in which a 10-week school-based intervention that focused on peer-to-peer mentoring was effective in enhancing self-regulation and engagement in physical activity among high school students. Allowing students to choose the exercises of their choice appears to have a positive effect on adherence to physical activities, which is expected to improve physical and mental health [22]. This was supported in a qualitative study by [22] as increased opportunities to participate in unstructured activities were key recommendations ofmost adolescent participants in the focus group discussion. Similarly, recorded significant improvements in physical activity among adolescents following centre-based childcare interventions [46]. Overall, the significant increase in physical activity after the intervention has vital health implications. In addition to bone health, which requires high-impact weight-bearing sports, aerobic exercises that stimulate the heart and lungs should provide the greatest health advantages [27,47]. The number of push-ups for 1 min, the number of sit ups for 1 min, and strength of hand recorded in the present study are considered high-impact aerobic exercises with a positive effect on the conditions of the lungs and heart.

The intervention in the present study was delivered by experts and driven by a foundation derived from the eight principles of PYD: sports skills, school engagement, healthy living, positive character, self-direction, teamwork, leadership, and community engagement. Previous studies reporting interventions that were developed based on some of these principles have also recorded a remarkable improvement in physical activity. For instance, interventions encompassing structured activities delivered by experts and that were theory-driven were associated with a moderate to high increase in physical activity among children and students [23].

Given the significant improvement in physical activities, particularly in the intervention group, the body fat composition in the group declinedsignificantly one month after the intervention. As observed in this study, students who participated in school-based programmes including physical activities experienced a significant reduction in BMI [42,48]. These studies entailed the use of multicomponent interventions comprising components related to physical activity and diet. Similarly, the mentorship programme used in the current study is best described as a multi-component intervention which entailed physical activities and brief discussions to motivate the students toward positive behavioural change. A systematic review conducted by [39] also concluded that interventions to enhance healthy nutrition and physical activity, including mentorship programmes, had positive impacts on BMI.

The reduction in the body fat composition in the intervention group implies potential benefits for participants. For instance, previous studies have demonstrated the link between the levels of cardiorespiratory fitness and abdominal adiposity, while the former is associated with evolving risk factors for cardiovascular diseases [27,29]. Moreover, obesity and increased body fat composition are established risk factors for cardiovascular disorders. Atherosclerosis develops earlier as childhood obesity rises. To address this public health issue, it is criticaltoemphasisegood lifestyle choices and regular exercise, especially among young people. Regular exercise may directly impact systemic circulation, improve insulin and adrenalin sensitivity, boost non-insulin-dependent glucose absorption, and enhance oxidative enzymes involved in carbohydrate and fat metabolism [45]. These events may explain the positive impacts of the mentorship programmeinvestigated in this study on body fat composition.

Another important aspect explored in this study is the participants’ relationship with family members and schoolmates post-intervention. Those in the intervention group experienced a significant increase in their relationship with family members. This result is not surprising given the indications of improved physical and mental health, as discussed in the aforementioned sections. A few studies have shown that significant improvement in students’ physical and mental well-being can be reflected in their study performance and interaction with co-workers or family relatives [31,49].

The present study depicts that school students’ post-intervention life skills scores increased by employing a module for the intervention of education and collaborative teaching-learning techniques. In the phase after the intervention, higher scores were seen in most life skill domains. Including this modular life skills training in education, the curriculum will foster their personality developmentand equip them to tackle various obstacles in life. In addition, the current study demonstrated that Chinese teenage mental health and PA levels are significantly positively correlated.

## 5. Conclusions

This study revealed the effectiveness of an intervention based on a mentorship programme comprising sport-based development and discussion on mental health and physical fitness. All of the physical activities and specific exercises were significantly enhanced in the intervention group compared to the control group. Indicators of physical and mental health such as self-efficacy, psychological resilience, and body fat composition were also substantially improved in the intervention group. These findings were reflected in the students’ relationships with their family members. A similarstudy should be conducted atother universities to elucidate the interventional programs and modify them to suit the general students.

## Figures and Tables

**Figure 1 ijerph-20-03737-f001:**
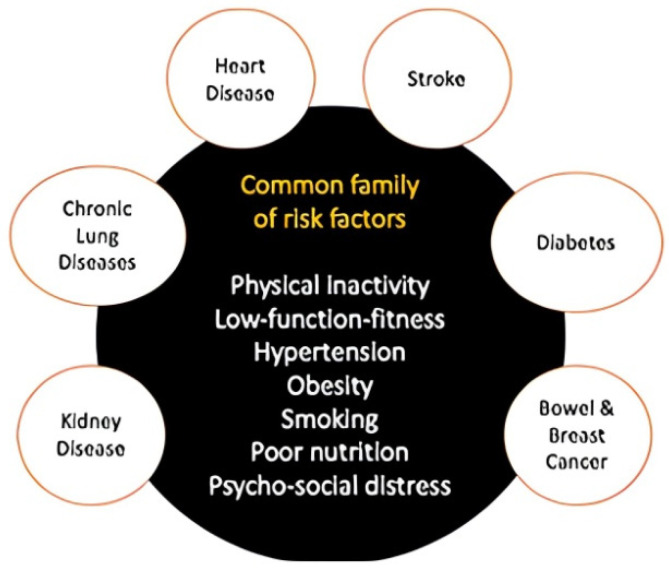
The consequences of physical inactivity [9]. (Source: Buckley, 2021).

**Table 1 ijerph-20-03737-t001:** Participants’ baseline characteristics.

Characteristics	Control Group *n* = 236	Intervention Group *n* = 194	*p*-Value *
Age (years, mean ± SD)	22.52 ± 3.60	22.29 ± 3.15	0.122
Gender (M/F)	(119/117)	(94/100)	0.241
BMI	26.95 ± 2.90	26.65 ± 2.69	0.075
Socio-economic class			
Lower	78	71	0.085
Middle	81	56	0.096
Higher	77	67	0.061

Notes: BMI = Body mass index; F = female; M = male; SD = standard deviation; * showingthesignificance of the difference between the control and intervention group; level of significance, α = 0.05.

**Table 2 ijerph-20-03737-t002:** Comparisons of overall physical activity and specific exercises performed in the intervention and control groups.

Parameters	Control		Intervention	
	Before	After	Before	After
	Mean ±SD	Mean ±SD	Mean ±SD	Mean ±SD
Number of push-ups for 1 min	21.91 ± 6.17	20.33 ± 5.20	20.59 ± 6.03	28.65 ± 7.02 *
Number of Sit and ups test for 1 min	29.69 ± 5.66	30.22 ± 5.14	25.87 ± 3.72	32.66 ± 4.55 *
Strength of hand grip (kg)	40.18 ± 6.28	42.65 ± 6.35	44.85 ± 3.86	48.55 ± 4.65 *
Jump test while standing (cm)	128.68 ± 12.96	131.98 ± 11.54	130.09 ± 12.68	142.25 ± 13.41 *
Physical Activity	5.46 ± 1.38	6.12 ± 1.33	5.48 ± 1.37	8.56 ± 1.21 *

Notes: SD = standard deviation; * mean values are statistically different compared to the baseline or before the intervention.

**Table 3 ijerph-20-03737-t003:** Comparisons of body fat composition and psychological resilience between the intervention and control groups.

Parameters	Control		Intervention	
	Before	After	Before	After
	Mean ±SD	Mean ±SD	Mean ±SD	Mean ±SD
Body Fat Proportion	21.40 ± 2.57	22.61 ± 2.80	19.3 ± 4.10	16.22 ± 2.30 *
Psychological Resilience	64.11 ± 10.77	65.31 ± 9.50	62.91 ± 10.92	69.85 ± 11.41 *

Notes: SD = standard deviation; * mean values are statistically different compared to the baseline or before the intervention.

**Table 4 ijerph-20-03737-t004:** Comparisons of self-efficacy and relationship with family and school activities between the intervention and control groups.

Parameters	Control		Intervention	
	Before	After	Before	After
	Mean ±SD	Mean ±SD	Mean ±SD	Mean ±SD
Self-efficacy	27.89 ± 4.00	28.41 ± 3.69	27.73 ± 3.79	31.25 ± 2.90 *
Relationship with family	42.13 ± 5.21	43.95 ± 5.33	41.61 ± 5.13	44.11 ± 5.10 *
Relationship with school	24.86 ± 2.34	25.62 ± 3.10	24.14 ± 2.75	25.85 ± 2.94

Notes: SD = standard deviation; * mean values are statistically different compared to the baseline or before the intervention.

## Data Availability

The data presented in this study are available on request from the corresponding author. The data are not publicly available due to privacy.

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
