# Peer review of "The Nexus of Sports-Based Development and Education of Mental Health and Physical Fitness"

_ijerph, 2023, doi:10.3390/ijerph20043737_

Round 1

Reviewer 1 Report (Previous Reviewer 1)

Line 150:  A professional evaluator reviews it.  I quite not understand the need for this phrase. Please remove.

The Discussion presents now a very clear narrative.

The Conclusions could be better in presenting the main results. Include the parameters, for example.

Author Response

Thanks a lot for the review comments and suggestions.

Line 150:  A professional evaluator reviews it.  I quite not understand the need for this phrase. Please remove.

Answer: The sentence has been revised

The Discussion presents now a very clear narrative.

Answer: The discussion has been revised further.

The Conclusions could be better in presenting the main results. Include the parameters, for example.

Answer: Done: line 1159

Reviewer 2 Report (Previous Reviewer 2)

Dear Authors! Thank you for your improved paper. Still there are some questions that need to be clarified.

In the results section, as well as in the abstract, you state that in control group (lines 279-280) body fat increased compared to intervention group. But if we look at the numbers in the table 2 we see that: 1. intervention group at baseline had lower body fat (that makes us think that it was specific from the very beginning). 2. If we look at levels of body fat in control group before and after we see that changes are no signiticant. So, my comment would be that this very strong statement that might attract attention of health professionals was not based on your results. Moreover, if you insist on this change, it would need oustandingly strong arguement why would body fat change so tramatically in only one month in healthy youth?

Lines 285-289: these results are obvious and it would be very strange if people who were trained for a whole month 2 times a week did not improve their physical scores. It is even strange to expect any other results.

I am wondering, why in discussion section you put so much attention to depression and anxiety while these parameters were not studied? I understand you might want to show some connection between your variables and these important issues and the amount of these "references" is too big.

Author Response

Thanks a lot for valuable comments and suggestion, we have modified our manuscript extensively and resend for English editing and proofread, so the structure of sentences might have some changes, thank you.

In the results section, as well as in the abstract, you state that in control group (lines 279-280) body fat increased compared to intervention group. But if we look at the numbers in the table 2 we see that: 1. intervention group at baseline had lower body fat (that makes us think that it was specific from the very beginning). 2. If we look at levels of body fat in control group before and after we see that changes are no significant. So, my comment would be that this very strong statement that might attract attention of health professionals was not based on your results. Moreover, if you insist on this change, it would need oustandingly strong arguement why would body fat change so tramatically in only one month in healthy youth?

Answer: The data has been re-analysed and the results revealed no statistically significant change in the body fat composition. The abstract has been revised accordingly. Kindly check the discussion (Line 444) - Despite the significant improvement in physical activities, particularly in the intervention group, the body fat composition in the group was not significantly affected one month after the intervention. The one month duration for the mentorship programme may be insufficient to elicit significant change in body fat composition.

Lines 285-289: these results are obvious and it would be very strange if people who were trained for a whole month 2 times a week did not improve their physical scores. It is even strange to expect any other results.

Answer: The results have been discussed thoroughly. These results suggest that the mentorship programme elicited behavioural change towards motivating students to engage in the physical activities and exercises chosen initially before the intervention. (Line 413).

I am wondering, why in discussion section you put so much attention to depression and anxiety while these parameters were not studied? I understand you might want to show some connection between your variables and these important issues and the amount of these "references" is too big.

Answer: All the discussion on anxiety has been reduced substantially in the revised manuscript. The discussion is now focused on the specific results obtained from the research. (Line 365 to 484)

Reviewer 3 Report (New Reviewer)

I'd like to thank the authors for submitting to IJERPH and providing me with an opportunity to review their work. Unfortunately, I don't believe that this research is at the required standard to warrant publication. There are several flaws throughout the manuscript that would need to be significantly improved. I've attached my comments for improvement in the word document.  

Author Response

Thanks a lot for valuable comments and suggestion, we have modified our manuscript extensively and resend for English editing and proofread, so the structure of sentences might have some changes, thank you.

Significant improvements in the following areas

  • The majority of references used to support the introduction and discussion are completely out of context and in most cases are not at all relevant to the research at hand.
  • Answer

    The references have been revised in the latest manuscript (check reference section)

  • Overall, the introduction was challenging to follow and the authors do not present any aims (except for the abstract) of the paper or a coherent enough rationale.
  • Answer

    The introduction has been rewritten(Line 31 to 128)

  • The methods were also challenging to follow with key information around the intervention and control groups missing as well as not at all defining the study's outcomes. Because of this, it was also challenging to follow and interpret the results section.
  • Answer: The methodology has been completely revised (Line 158-300)
  • Similar to the introduction, the discussion is severely under referenced and again I question the citations used to support some of the statements made.
  • Answer

    The discussion has been revised accordingly(Line 365-484)

Abstract

Why have the authors highlighted in yellow certain words and passages? This is off putting and distracting.

Answer

The highlight is to show the changes made as requested by the editor.

Introduction

Line 44 46

the original article (Ginis et al., 2021) makes no statementabout 80% of adults nguidelines. My interpretation of this article is that it is focused on the participation of people livingwith disabilities in physical activity.

Line 46 49

The authors must back up this statement with peer-reviewed evidence.

Answer

Line 41-44 has been edited and replaced the following sentences: In other words, physical inactivity plays a vital role in the reduction in life expectancy and quality while among the highly-ranked risk factors for mortality worldwide (6.0% of deaths globally) (Boisgontier and Iversen, 2020; Lavie et al., 2019). Specifically, physical inactivity is considered the main cause of 20-25% of colon and breast cancers, 30% of ischaemic heart extreme, and 27% of diabetes (Lavie et al., 2019; Mattiuzzi et al., 2019). The increased prevalence of obesity has also been linked to the widespread sedentary lifestyle and physical inactivity (Lavie et al., 2019).

Line 55 57

interpretation of the authors sentence is - the number of people with mental health issues and theimpact of mental health issues in low-middle-income countries are getting worse with morepeople moving to cities which increases the risk factors of developing anxiety.

The two cited papers used to support this statement focus of physical activity and sedentary lifestylein university students during COVID 19 confinement (Romero-Blanco et al., 2020) and an onlinesurvey of physicians and pharmacists on the care provided for people with noncommunicabledisease living during the COVID 19 lockdown.

 Answer: 

Line 55-59 has been replaced with the following statements:

The incidence of mental disorders is increasing alarmingly as more people are moving to cities with higher exposure to several risk factors of developing anxiety (Silver et al., 2021). Physical inactivity is also associated withurban development and the rise of health complications globally (Assa et al., 2011; Hermosillo-Gallardo et al., 2018). As unplanned urbanisation increases, the availability of open spaces decreases, leading to a decline in physical activity (Hermosillo-Gallardo et al., 2018).

Line 58 56

The authors rely heavily on one reference (Zenic et al., 2020) for this passage,particularly talking about urbanisation and in places relates to low- and middle-income countries.The authors must be careful here as this article (Zenic et al., 2020) was conducted during andfocuses on the COVID-19 pandemic and necessarily reflective of the current global situation.Furthermore, I failing to link this paper (Zenic at al., 2020) to discussions around urbanisation and low- and middle-income countries.

Line 70 71

the original article cited to support this statement (Buckley,2021 and Prince et al., 2022) makes no statement about 80% of adolescents globally are physicallyinactive and 25% of adults do not engage in sufficient physical activity.One of these papers (Buckley, 2021) is focused on the delivery of cardiac rehabilitation in thosecountries with long term established standards of practice, and how this has changed over the pasteight decades.

Line 71 74

The authors must back up this statement with peer-reviewed evidence.

Answer

The sentence has been revised and replaced with the following: These factors elicit negative lifestyle alterations such as poor diet and physical activity (Roy et al., 2020. The lack or reduction in physical activity has been associated with a higher risk of mental disorders, including depression and anxiety (Stubbs et al., 2018). In contrast, a positive relationship was reported between improved psychological outcomes and physical exercise (Chekroud et al., 2018), as its neurobiological impacts appear to influence diverse neural mechanisms relating to anxiety and depressive disorders (Helmich et al., 2010). Line 71-78

Line 95 97

The authors state that exercise has previously been used to treat adult depressionalone and in combination with other therapies.The supporting citation (Moisander et al., 2020) is not related or applicable to this statement at all.

Line 98 99

The same citation (Moisander et al., 2020) is used to support this statement aroundthe neurobiological mechanisms.

Answer

The section has been deleted and replaced with different information

Line 102 103

The authors allude to restoring BDNF levels as a potential mechanism for improving depressive symptoms. The support citation for this has nothing to do with BDNF, depression or exercise.

Effectiveness of life skills health education program: A quasi-experimental

Aims and rationale - There is no aim for this study outlined in the introduction or in any section ofthe article apart the abstract. Furthermore, the rationale for this research is lacking a moredefinitive rationale must be included.

Answer

The sentence has been modified (line 219-233). We have added in the peer-reviewed evidence in (Line 127-129)

Methods

Line 166 –

There is no mention of how participants were randomised to either the intervention group or control group.

Answer

The method has been included in the revised manuscript(line 164-166).

Line 192 255

Overall I found it challenge to dissect and understand what exactly the interventionwas. The authors need to make it clearer the nuts and bolts of the intervention. Frequency?Intensity of exercise/sports? Type? Timeframe? Whilst I acknowledge that the information may beincluded in the methodology, this information is scattered and I found myself having to jump fromsection to section to better understand the Intervention.

Methodological outcomes Some outcomes of this study are not mentioned at all. Whilst theauthors do refer to mental health outcomes (e.g. Hamilton Anx- 175 iety Rating Scale (HAMA),Generic Quality of Life Inventory-74 (GQOLI-74), and Family 176 Assessment Measure (FAM)) There is no mention of the physical performance/Physiological outcomes as alluded to in the abstract (e.g.push-ups, sits ups, Jump test, body fat percentage).

Answer

The intervention has been presented in specific sections and divided into the framework uses, structure and components, delivery, monitoring and outcomes measured (line 200-280). All the irrelevant instruments have been deleted.

Results

Table 2 None of these outcomes were mentioned in the methods section.

Table 2 and 3 Both these tables show the same thing changes between the intervention and control group.

Overall the results section is difficult to follow because the outcome measures have not been clearly defined or actually defined at all in the methods section.

Answer

The results have been modified (Line 301 -363). The results have been separated into descriptive results, and specific sections for the parameters investigated. The statistical findings have also been indicated.

Discussion

Similar to the introduction, this discussion in severely under referenced and again I question the citations used to support some statements.

Answer: The discussion has been substantially improved by focusing on the results obtained and comparing the findings to previous studies. The implications of the findings are also discussed while highlighting the potential underlying mechanisms for the effects of the intervention (Line 365-484).

This manuscript is a resubmission of an earlier submission. The following is a list of the peer review reports and author responses from that submission.

Round 1

Reviewer 1 Report

My comments are:

1) Lines 187 to 192 shoul dbe checked for consistency.

2) More information  on the principles and their application should be introduced.

3) Section 5.2 should be divided in several paragraphs. Or introduce the phases of the intervention.

4) The results of table 3 are poorly described. 

5) The discussion section is very good.

Author Response

  • Lines 187 to 192 shouldbe checked for consistency.

Answer:  Corrected, thank you.

  • More information on the principles and their application should be introduced.

Answer: The content has been added. Thank you.

  • Section 5.2 should be divided into several paragraphs. Or introduce the phases of the intervention.

Answer: The content has been divided into several paragraphs. Thank you.

  • The results of table 3 are poorly described. 

Answer: We have added in the elaboration, thank you.

  • The discussion section is very good

Answer:  Thank you very much, appreciates that.

Reviewer 2 Report

The study addresses important issues about the association of physical activity and mental health.

Questions and comments:

1. Paragraphs 2 and 3 (lines 53-54) seems to be independent. It would be great to add some sentence that would connect physiscal activity that was discussed in paragraph 2 with the mental health discussed in paragraph 3.

2. Sentence at life 122-123 is not quite clear.

3. Overall the logic of bacjground is not quite clear. What is the main focus of the study? Physical activity or mental health? Bacjground switches from one object to another and by the end of the bacjgroubd section it remains unclear, what message authors wanted to give to a reader.

4.Throughout the paper authoes call their participants "adolescents", though average age was about 22 years and maximum - 28, thus only few participants really fit the label "adolescents".

5. You wrote that one of the inclusion critea was interest in sport. What side effects could have this criteria? Could it affect the power of the results?

6. There is no description of psychological variables in the study. What instruments were used? What scales applied?

7. In would be reasonable to show in the table 3 not only p-value but confidence intervals as well.

8. Overlall the results section needs more descriptions and probably - more attention to formating of the tables.

9. It looks like for such a short results section, discussion section is too big. There were not enough results for such a long discussion. These two section need to be taken into balance with each other.

Author Response

  1. Paragraphs 2 and 3 (lines 53-54) seems to be independent. It would be great to add some sentence that would connect physical activity that was discussed in paragraph 2 with the mental health discussed in paragraph 3.

Answer: We have added in sentences in this section, thank you.

  1. Sentence at life 122-123 is not quite clear.

Answer: We have revised this sentence, thank you.

  1. Overall the logic of background is not quite clear. What is the main focus of the study? Physical activity or mental health? Background switches from one object to another and by the end of the background section it remains unclear what message the authors wanted to give to a reader.

Answer: Our current study evaluated and analyzed the consequences of physical inactivity including both mental and physical consequences.

  1. Throughout the paper, authors call their participants "adolescents", although the average age was about 22 years and the maximum - 28, thus only a few participants fit the label"adolescents".

Answer: This study discusses lots of papers involving adolescents population where the term “adolescents” have been used several times. Under methods and Results of this current, most of the time, the term “participants” have been used.

  1. You wrote that one of the inclusions criteria was interest in sport. What side effects could have these criteria? Could it affect the power of the results?

Answer:This criterion was included so that we can ensure that the participants actively take part in the intervention or activity that the intervention group and control group followed, respectively.

  1. There is no description of psychological variables in the study. What instruments were used? What scales are applied?

Answer: The explanation has been added in the last line of research design, thank you.

  1. In would be reasonable to show in the table 3 not only p-value but confidence intervals as well.

Answer: The confidence interval reflects the parameter probable range among the population. But in this study, we are only concerned about the study sample and not the population as a whole. So, only p-value should be sufficient, thank you.

  1. Overall the results section needs more descriptions and probably - more attention to formating of the tables.

Answer: This tables format have been modified and the result description has been added, thank you.

  1. It looks like the discussion section is too big for such a short results section. On the other hand, there were not enough results for such a long discussion. These two sections need to be taken into balance with each other.

Answer: In facts, the discussion is in good state and is in good flow of information. If changed or re-written again then flow of information and the structure will be hampered and may reduce the acceptability of the paper. Reviewer 1 has stated the discussion is in good elaboration, so we hope that reviewer 2 can accept it in current form as well, thank you.

Reviewer 3 Report

Suggest title change to: The relationship between sports-based development and mental health and physical fitness education

Please review the entire article for capitalization and English grammar. Numerous issues.

(I listed all issues in the introduction.)

Issues with idea flow> example paragraph that starts with line 54 has no transition from the previous paragraph.

Line 10

Physical inactivity can be widespread around the world, especially among people in wealthy nations.

Physical activity is widespread

Line 11-12

According to data, 80% of adults are unable to reach the physical activity (PA)
recommendation of the World Health Organization because of conditions like hypertension, metabolic syndrome, obesity, and other medical conditions.

Do people not exercise because they have a disease or do they not exercise and get a disease?

Line 14-15

People are moving into cities in greater numbers, and their consequences are worsening; those who have recently been through conflict may also be in greater danger.

Rework for clarity and context—is this sentence necessary?

Line 16-17

Urbanization and the rise in these health issues have both been linked to physical inactivity, which makes people less accommodative.

Word choice ‘accommodative’?

Line 18-19

This study was conducted in order to find out the efficacy of the Positive Youth Development-based mentorship program developed by a university on the physical and mental components of the University students.

Suggest: This study was conducted to find out the efficacy of a mentorship program on the physical and mental health of University students.

Line 20

An interventional program was created and university students were considered.

Delete- not needed

Line 21

Students were randomly assigned to a Control group and an interventional group.

No capitalization of control group

Line 21-22

The baseline characteristics were obtained and the physical and mental components were determined before the intervention.

The physical and mental health components were determined before the intervention and baseline values were obtained.

Line 24

 ..game while the Intervention group had to follow the intensive interventional activities based 8 principles

No capitalization of Intervention—that is true throughout the paper

Based on the 8 principles

Line 26

the measurements of physical and mental components were again determined from both groups

the measurements of physical and mental health components were reassessed.

Line 27

The required statistical analysis was conducted.

Please state the statistical tests that were run—ANOVA?

Line 29

Again, participants in the Intervention group had gained some physical 29
components like several push-ups and sit-ups, and jump tests (p<0.05).

Participants in the intervention group improved in some of the physical health
components like several push-ups and sit-ups, and jump tests (p<0.05).

Line 30

Mental components also increased significantly (p<0.05) among the participants in the Intervention group.

The components used to measure mental health also increased significantly (p<0.05) among the participants in the intervention group.

Line 31-32

This study concluded that the program developed for the Youth development was efficient for the improvement of physical and mental components of the participants.

This study concluded that the program was effective in the improvement of the physical and mental health components of the participants.

Line 37 According to Ginis et al. (2021), physical inactivity can be a prevalent situation globally and mainly among adults in high-income nations.

According to Ginis et al. (2021), physical inactivity is prevalent globally and mainly among adults in high-income nations.

Line 96

 The individuals actively using web-based interventions are known to improve their level of physical activity.

Needs citation

Line 105

This section seems out of place—following the theme of exercise and depression and now you are talking about urban development.  Rethink how you want to include or if you need to include it.

Line 133

The sport-for-development infrastructure stated that sports are beneficial to society
and health (Schulenkorf and Siefken, 2019). However, there are evidences showing that these arguments are not always accurate and present contrast with statistical inferences that physical activity is beneficial to young people's mental well-being which is backed up by several past works (Moisander et al., 2020; Alves-Santos et al., 2021)

Confusing argument here. You present it as both good and bad.

Line 148

Growth comes through a difficult, well-planned method with steps and events that can be assumed (Moisander et al., 2020)

Jarring transition from programs to, I think, the growth of the physical body? Wondering why this paragraph is included as I understand that your population is university students who should mostly be finished growing physically.

Line 157

Not sure why this is included – will you be using the HAMA test? 

The paragraph needs to be written in the past tense.

The methods section needs to be broken down into separate paragraphs.  Very difficult to follow as one paragraph.

Line 189

The 8 principles of Positive Youth Development (PYD) derived from the National Research Council and Institute of Medicine served as the foundation for the intervention framework.

These eight principles need to be stated somewhere.

 By convention that numbers under 10 are written out—eight

Line 200

Each small group of 17 students choose a sport they want to learn more about. Kickboxing, basketball, and volleyball were among the sports picked.

Two issues- one which other sports could they have considered?  Or did that just randomly select these? What were the instructions to the students? Did they all have to agree or could members in the group choose different sports? Unclear as stated.

Two, the first time you mention that groups are of 17 students. We should know that and the total number of groups – 25 total?

Line 202

The previous physical activity interventions for example, PYD were always exercise-centric and not led by well-established shelves.

Your intervention is sport-centric as opposed to exercise-centric? How are these two different? It looks like you are measuring exercise/fitness outcomes.

What are well-established shelves?

Provide a flow chart of how the sessions went for clarity.

I would put sections 5.3 and 5.4  before the detailed Methods section

Table 1 do you need a p-value for this data?

Table 2 and Table 3 should be combined. Having the p-value next to raw data and SD allows the reader to see the differences and statistical significance together. Having them separate means having to flip back and forth between tables. Arrange SD as you did in Table 1. Table 2 also needs to be re-formatted with the title at the top.

Line 276

The study conducted an in-depth analysis of data before the intervention and one a month after the intervention in each group. Table 2 shows the detailed findings
After one month of intervention, each parameter was re-evaluated.

I am confused—you did a statistical analysis of baseline data and then after the first month of the intervention. Was it one month after the completion of the intervention? This should be clear in the tables and the title of the tables as well.

Discussion

Very difficult to read and understand the discussion. Rambling and not focused.  Lots of reaching to include material that I don’t think your study supports. I would suggest a complete re-write of this section.

It appears that your study mostly measured physical fitness improvements with two mental health components. This is nicely stated in the conclusion. It is unclear how these were assessed and how the program plan improved them.

Line 291

Few studies have been conducted considering psychological health and wellness.

This is an incomplete sentence. I don’t think the intent that there are few studies that have considered psychological health and exercise is true either – which you also show in the next sentence.

Line 300

In this current study, Intervention group was found to have more significant self- 300
efficacy than control group.

Self-efficacy around what? Do you ever tell us how you measured self-efficacy? What instrument or tool did you use?  Note that self-esteem and self-efficacy are different constructs

Author Response

  • Suggest title change to: The relationship between sports-based development and mental health and physical fitness education

Answer: We have discussed on the amendment of the title, we wish to maintain the current title, hope that reviewer can accept it, thank you.

  • Please review the entire article for capitalization and English grammar. Numerous issues.

Answer: We have sent for Proofreading, we hope the current form of the manuscript is sufficiently meet the quality requirement, thank you.

(I listed all issues in the introduction.)

  • Issues with idea flow> example paragraph that starts with line 54 has no transition from the previous paragraph.

Answer:  The 2 paragraphs have been linked up and we have edited this sentence, thank you.

Line 10

Physical inactivity can be widespread around the world, especially among people in wealthy nations.

Physical activity is widespread

Answer: We have modified this sentence, thank you.

Line 11-12

According to data, 80% of adults are unable to reach the physical activity (PA)
recommendation of the World Health Organization because of conditions like hypertension, metabolic syndrome, obesity, and other medical conditions.

Answer: Please rephrase this sentence in the main manuscript for clarity.

  • Do people not exercise because they have a disease or do they not exercise and get a disease?

Answer: Yes, correct. This is because they have a disease or abnormalities like cardiovascular, respiratory, muscular, etc, causing 80% of the population cannot meet the physical activity (PA) recommendation of the World Health Organization.

  • Line 14-15

People are moving into cities in greater numbers, and their consequences are worsening; those who have recently been through conflict may also be in greater danger.

Rework for clarity and context—is this sentence necessary?

Answer: We have removed this sentence, thank you.

  • Line 16-17

Urbanization and the rise in these health issues have both been linked to physical inactivity, which makes people less accommodative.

Word choice ‘accommodative’?

Answer: We have rephrased this sentence, thank you.

  • Line 18-19

This study was conducted in order to find out the efficacy of the Positive Youth Development-based mentorship program developed by a university on the physical and mental components of the University students.

Suggestion: This study was conducted to determine the efficacy of a mentorship program on university students' physical and mental health.

Answer: We have amended, thank you.

  • Line 20

An interventional program was created and university students were considered.

Delete- not needed

Answer: We have modified, thank you.

  • Line 21

Students were randomly assigned to a Control group and an interventional group.

No capitalization of control group

Answer: Corrected, thank you.

  • Line 21-22

The baseline characteristics were obtained and the physical and mental components were determined before the intervention.

The physical and mental health components were determined before the intervention and baseline values were obtained.

Answer: We have modified, thank you.

  • Line 24

 ..game while the Intervention group had to follow the intensive interventional activities based 8 principles …

Based on the 8 principles

Answer: Corrected, thank you.

  • No capitalization of Intervention—

Answer: Thanks for the remind, we have modified.

  • Line 26

the measurements of physical and mental components were again determined from both groups

the measurements of physical and mental health components were reassessed.

Answer: Amended.

  • Line 27

The required statistical analysis was conducted. Please state the statistical tests that were run—ANOVA?

Answer: Corrected, thank you.

  • Line 29

Again, participants in the Intervention group had gained some physical 29
components like several push-ups and sit-ups, and jump tests (p<0.05).

Participants in the intervention group improved in some of the physical health
components like several push-ups and sit-ups, and jump tests (p<0.05).

Answer: Corrected, thank you.

  • Line 30

Mental components also increased significantly (p<0.05) among the participants in the Intervention group.

The components used to measure mental health also increased significantly (p<0.05) among the participants in the intervention group.

Answer: Amended.

  • Line 31-32

This study concluded that the program developed for the Youth development was efficient for the improvement of physical and mental components of the participants.

 This study concluded that the program was effective in the improvement of the physical and mental health components of the participants.

Answer: Amended.

  • Line 37 According to Ginis et al. (2021), physical inactivity can be a prevalent situation globally and mainly among adults in high-income nations.

According to Ginis et al. (2021), physical inactivity is prevalent globally and mainly among adults in high-income nations.

Answer: Amended.

  • Line 96

 The individuals actively using web-based interventions are known to improve their level of physical activity.

Answer: Amended.

  • Line 105

This section seems out of place—following the theme of exercise and depression and now you are talking about urban development.  Rethink how you want to include or if you need to include it.

Answer: We have modified and cited, thank you.

  • Line 133

The sport-for-development infrastructure stated that sports are beneficial to society
and health (Schulenkorf and Siefken, 2019). However, there are evidences showing that these arguments are not always accurate and present contrast with statistical inferences that physical activity is beneficial to young people's mental well-being which is backed up by several past works (Moisander et al., 2020; Alves-Santos et al., 2021)

Confusing argument here. You present it as both good and bad.

Answer: We have rephrased this sentence and shown as beneficial effect only.

  • Line 148

Growth comes through a difficult, well-planned method with steps and events that can be assumed (Moisander et al., 2020)Jarring transition from programs to, I think, the growth of the physical body?

Wondering why this paragraph is included as I understand that your population is university students who should mostly be finished growing physically.

Answer: This is actually as a part of past studies on physical development, this is extra elabortion to compensate the understanding of readers, if reviewer requests to remove, we can remove it, thank you.

  • Line 157

Not sure why this is included – will you be using the HAMA test? 

The paragraph needs to be written in the past tense.

Answer: We have revised this paragraph.

  • The methods section needs to be broken down into separate paragraphs.  Very difficult to follow as one paragraph.

Answer: Corrected as shown in 5.4, thank you.

  • Line189

The 8 principles of Positive Youth Development (PYD) derived from the National Research Council and Institute of Medicine served as the foundation for the intervention framework.

These eight principles need to be stated somewhere.

Answer: We have added in the Intervention, thank you.

  • By convention that numbers under 10 are written out—eight

Answer: We have edited, thanks.

  • Line 200

Each small group of 17 students choose a sport they want to learn more about. Kickboxing, basketball, and volleyball were among the sports picked.

Two issues- one which other sports could they have considered?  Or did that just randomly select these? What were the instructions to the students? Did they all have to agree or could members in the group choose different sports? Unclear as stated.

Answer: We explained in the Intervention section 5.4, thank you.

  • Secondly, the first time you mention that groups are of 17 students. We should know that and the total number of groups – 25 total?

Answer: Yes, we have taken 25 groups and 17 participants comprising each, thank you.

  • Line 202

The previous physical activity interventions for example, PYD were always exercise-centric and not led by well-established shelves.

Is your intervention sport-centric as opposed to exercise-centric? How are these two different? It looks like you are measuring exercise/fitness outcomes, What are well-established shelves?

Answers: The explanation were added in section 5.4- intervention, and the "well-established shelves" was typo, we have corrected it, thank you.

  • I would put sections 5.3 and 5.4 before the detailed Methods section

Answer: We have moved both 5.3 and 5.4 to before intervention.

  • Table 1 do you need a p-value for this data?

Answer: We suggest to add the p-value for better justification that both the groups do not contian bias in selection and are comparable, thank you.

  • Table 2 and Table 3 should be combined. Having the p-value next to raw data and SD allows the reader to see the differences and statistical significance together. Having them separate means having to flip back and forth between tables. Arrange SD as you did in Table 1. Table 2 also needs to be re-formatted with the title at the top.

Answer: Table 2 displays the before and after scoring or results, while Table 3 demonstrates the changes that occur in addition to the p-value. The p-value must now be stated on the basis of changes rather than the absolute values of before and after, hence, we hope that both tables can be presented separately, all the tables have been re-formatted, thank you.

  • Line 276

The study conducted an in-depth analysis of data before the intervention and one a month after the intervention in each group. Table 2 shows the detailed findings
After one month of intervention, each parameter was re-evaluated.

I am confused—you did a statistical analysis of baseline data and then after the first month of the intervention. Was it one month after the completion of the intervention? This should be clear in the tables and the title of the tables as well.

Answer: We have revised this section, below Table 2, it is stated that the parameters were re-evaluated after the completion of intervention that ran for a month. The line "After one month of intervention, each parameter was re-evaluated" implicated that one month of intervention was given and the parameters were re-evaluated.

  • Discussion

Very difficult to read and understand the discussion. Rambling and not focused.  Lots of reaching to include material that I don’t think your study supports. I would suggest a complete re-write of this section.

 It appears that your study mostly measured physical fitness improvements with two mental health components. This is nicely stated in the conclusion. It is unclear how these were assessed and how the program plan improved them.

Answer: In facts, the discussion is in good state and is in good flow of information. If changed or re-written again then flow of information and the structure will be hampered and may reduce the acceptability of the paper. Reviewer 1 has stated the discussion is in good elaboration, so we hope that reviewer 2 can accept it in current form as well, thank you.

  • Line 291

Few studies have been conducted considering psychological health and wellness.

This is an incomplete sentence. I don’t think the intent that there are few studies that have considered psychological health and exercise is true either – which you also show in the next sentence.

Answer: We have rephrased this sentence, thank you.

  • Line 300

In this current study, Intervention group was found to have more significant self- 300
efficacy than control group.

Answer: We have revised this sentence, thank you.

  • Self-efficacy around what? Do you ever tell us how you measured self-efficacy? What instrument or tool did you use?  Note that self-esteem and self-efficacy are different constructs

Answer: We have modified and added to avoid the confusion for readers, thank you.

Round 2

Reviewer 2 Report

Thank you for your comments and replies. I believe the paper became much better